# Synthesis and Characterization of Environmentally Friendly Chitosan–Arabic Gum Nanoparticles for Encapsulation of Oregano Essential Oil in Pickering Emulsion

**DOI:** 10.3390/nano13192651

**Published:** 2023-09-26

**Authors:** Ismael Lobato-Guarnido, Germán Luzón, Francisco Ríos, Mercedes Fernández-Serrano

**Affiliations:** Department of Chemical Engineering, University of Granada, 18071 Granada, Spain; rios@ugr.es (F.R.); mferse@ugr.es (M.F.-S.)

**Keywords:** encapsulation, essential oil, Pickering emulsions, chitosan, Arabic gum, detergent formulation

## Abstract

The encapsulation of bioactive agents through the utilization of biodegradable nanoparticles is a topic of considerable scientific interest. In this study, microcapsules composed of chitosan (CS) and Arabic gum (GA) nanoparticles were synthesized, encapsulating oregano essential oil (OEO) through Pickering emulsions and subsequent spray drying. The optimization of hybrid chitosan and Arabic gum (CS–GA) nanoparticle formation was carried out via complex coacervation, followed by an assessment of their behavior during the formation of the emulsion. Measurements of the size, contact angle, and interfacial tension of the formed complexes were conducted to facilitate the development of Pickering emulsions for encapsulating the oil under the most favorable conditions. The chitosan–Arabic gum capsules were physically characterized using scanning electron microscopy and fitted to the Beerkan estimation of soil transfer (BEST) model to determine their size distribution. Finally, the OEO encapsulation efficiency was also determined. The optimum scenario was achieved with the CS–GA 1–2 capsules at a concentration of 2% wt, featuring a contact angle of 89.1 degrees, which is ideal for the formation of oil/water (O/W) emulsions. Capsules of approximately 2.5 μm were obtained, accompanied by an encapsulation efficiency of approximately 60%. In addition, the hybrid nanoparticles that were obtained showed high biodegradability. The data within our study will contribute fundamental insights into CS–GA nanoparticles, and the quantitatively analyzed outcomes presented in this study will hold utility for forthcoming applications in environmentally friendly detergent formulations.

## 1. Introduction

Arabic gum (GA) is a natural arabinogalactan protein-type polysaccharide that is extracted from the trunks and branches of *Acacia senegal* and *Acacia seyal* trees [1,2]. As an anionic polysaccharide (pKa = 2.2), Arabic gum is a favored material for microencapsulation due to its remarkable water solubility, low solution viscosity, effective surface activity, and emulsification capacity [3,4].

Chitosan (CS) is a linear polysaccharide that is obtained from the partial deacetylation of chitin, and is composed of randomly distributed units of (1–4) linked D-glucosamine and N-acetyl glucosamine. After some adequate modifications to its structure, it has been reported to be an efficient raw material for producing nanoparticles with technological benefits [5,6]. It carries a positive charge within an acidic environment owing to its pKa, which falls between 6.2 and 7.0. This leads to the protonation of amino groups in the polymer backbone and abundant hydroxyl groups [7,8]. However, the existence of non-protonated NH_2_ groups within chitosan (CS) leads to its restricted solubility within physiological scenarios, thereby restricting the range of its applications. As a result, a compelling necessity has arisen to explore structural modifications to it or combine it with other compounds to enhance its usability.

The aim of this research was to investigate the synergy of the properties of these two biopolymers in the production of mixed nanoparticles with enhanced properties, efficiently contributing to the stabilization of O/W Pickering emulsions [9,10], which are universally applied in the pharmacology, biomedicine, and food industries. In addition, Pickering emulsions can also be elaborately designed to improve the stability of bioactive substances [11].

The formation of the hybrid particles is accomplished through complex coacervation [12]. This is a phase separation phenomenon that occurs when two oppositely charged macromolecules interact in a solution to form a dense coacervate-rich phase and a dilute supernatant-rich phase. Thus, when chitosan and Arabic gum are mixed in a solution, they undergo complex coacervation due to their opposite charges. Chitosan’s amino groups and Arabic gum’s carboxylate groups are electrostatically attracted to form coacervate complexes [13]. These complexes have been used in many different applications, such as the adsorption of heavy metals [14] in medicine [15,16,17] or agriculture [18,19]. However, their primary application lies in serving as carriers or encapsulating agents for active substances, multiplying their use in several industries. Some examples include their use as carriers of antibiofilm agents [20], antioxidants [21], nanoencapsulation drugs [22], or, as was the purpose of this study, encapsulating essential oils [23,24].

Oregano essential oil (OEO) is extracted from the oregano plant, *Origanum vulgare*, and is renowned for its antifungal and antibacterial attributes [25,26]. It poses no harm to human health or the environment [27]. The oil’s major components are carvacrol, γ-terpinene, thymol, β-citronellol, 1,8-cineole, etc. [25], which exhibit potent antioxidant characteristics that can aid in shielding against toxins and combatting fungal infections. Thus, these properties make OEO an attractive compound for detergent formulations [27].

The objective of this study was to obtain environmentally friendly OEO capsules made from chitosan and gum Arabic nanoparticles through optimization of the emulsion conditions, and to examine their contributions to different detergent formulations. Thus, the most important parameters, such as particle size, contact angle, and interfacial tension, can be managed to achieve the optimal conditions for producing chitosan–Arabic gum (CS–GA) capsules. The selection of the most favorable conditions could influence the formation of the emulsion and, consequently, the subsequent acquisition of capsules via spray drying atomization.

## 2. Materials and Methods

### 2.1. Chemicals

All of the chemicals were at least analytical grade. Low-molecular-weight chitosan powder was purchased from Sigma-Aldrich (Product No. 448869, Darmstadt, Germany), with a deacetylation degree of 91.7% and MW 359 kDa; Arabic gum (Ph Eur, BP) was purchased from Merck (Darmstadt, Germany); glacial acetic acid (ACS 99.7%) was purchased from PanReac (Barcelona, Spain); sunflower oil was purchased from Mercadona S.A (València, Spain); and oregano essential oil (OEO) was acquired from Essenciales (Barcelona, Spain). All of the chemicals were used as received. Ultra-pure water with a resistivity of 18 MΩ cm^−1^, obtained from a Milli-Q analytical reagent-grade water purification system (Millipore, Burlington, MA, USA), was utilized for all the experimental procedures.

### 2.2. Methods

#### 2.2.1. Synthesis of Nanoparticles, Preparation, and Complex Coacervation

The chitosan–Arabic gum (CS–GA) hybrid nanoparticles were prepared via complex coacervation. Two aqueous solutions were prepared for each compound. At the desired concentration, the appropriate amount of chitosan was weighed and diluted in the required volume of a 0.5 N solution of acetic acid. The Arabic gum was directly diluted in water.

For the formation of the nanoparticles, the Arabic gum solution was slowly added to the chitosan solution using a syringe pump (ASP-030, Syrris Ltd., Royston, UK) at a flow rate of 1 mL/min. Different concentrations and polymer ratios were tested to study their influence on the measured variables.

#### 2.2.2. Size and Zeta Potential

A Zetasizer Ultra (Malver Panalytical Ltd., Malvern, UK) was used to measure the size and zeta potential. The size was measured using a dynamic light scattering (DLS) technique, while the zeta potential was measured by electrophoretic light scattering (ELS) [28]. The instrument’s settings were optimized automatically by means of ZS XPLORER software v. 3.2.1.11 (Malvern Panalytical Ltd., Malvern, UK). All of the experiments were performed at room temperature. The data were the average of at least three replicates.

The Zetasizer Ultra was used to study the behavior of the different CS–GA nanoparticles prepared with respect to pH. The starting point was always the original pH of the prepared CS–GA nanoparticle solutions, and the data were taken at ascending intervals with pH changes of 1 unit, to have sufficient data. In all cases, the final pH was 10. For this, a 0.5 M NaOH solution was used, which the equipment itself dispenses to reach the desired pH values. When the desired pH was reached, the Z-potential was measured.

#### 2.2.3. Interfacial Tension and Contact Angles

The interfacial tension between the aqueous solution containing the nanoparticles and sunflower oil, and the contact angles of the particles were measured using a KSV CAM 200 drop tensiometer. The interfacial tension was determined using the pendant drop method, fitting each image of the CAM 200 to the Young–Laplace equation, enabling the calculation of the surface and interfacial tensions.

For the contact angles, a thin film of an aqueous solution of nanoparticles was dried overnight over a glass slide. A drop of water was placed over the dried nanoparticles, and the shape of the drop placed was fitted to the Young–Laplace equation to determine the contact angle.

#### 2.2.4. Biodegradability Test

Aerobic biodegradation tests were conducted in accordance with the method of OECD 301F: Ready Biodegradability [29]. The OXITOP^®^-IDS system (v1.06) was used, which monitors biodegradation through indirect measurements of the O_2_ consumed. Activated sludge from the local wastewater treatment plant served as the source of microorganisms. The duration of the test was 28 days. The materials were classified as readily biodegradable if the percentage of biodegradation exceeded 60% within 28 days. Conversely, materials failing to meet these criteria were categorized as non-biodegradable. Microcrystalline cellulose served as the reference compound.

To determine the theoretical oxygen demand of the tested compounds, a preliminary elemental analysis was conducted to establish the optimal quantity of each sample required for the successful execution of the method.

All of the experiments were performed in duplicate.

#### 2.2.5. Encapsulation

First, emulsions of oregano essential oil in water were prepared. The aqueous phase consisted of a water solution of 2% wt of CS–GA nanoparticles prepared as described in Section 2.2.1. For the formation of the emulsion, a Silverson L5M (Silverson^®^, Chesham, UK) rotor–stator mixer equipped with a general-purpose disintegrating head was used. The first stage was to add Tween 80 (5%v) surfactant to the aqueous phase; agitation was maintained for 1 min at 5000 rpm. After that, OEO was slowly added, and three cycles of 60 s and 5000 rpm were applied to the mixture, with a 30 s rest between each cycle.

Encapsulation of the OEO was achieved by spray-drying using a laboratory-scale spray-drier (Büchi B-190; Büchi Labortechnik, Flawill, Switzerland) at 180/90 °C inlet/outlet temperatures, respectively. The flow of drying air was fixed to 25 Nm^3^/h.

The system functioned in monoaxial mode, introducing the formed emulsion to be dried, and obtaining capsules containing the oil in the collector of the equipment after passing through the cyclone.

The capsules were prepared using two different concentrations of OEO, 0.5%v and 10%v, in an emulsion with respect to the aqueous phase, with two very different values for analyzing how the amount of oil affected the formation of the capsules.

#### 2.2.6. Formation of the Capsules: Size and Morphology

The capsules’ morphology was examined via scanning electron microscopy (SEM), using an FEG-ESEM QUEMSCAN 650F microscope. Depending on the microencapsulation technique used, a thin layer of microcapsules was positioned on a carbon tape and coated with carbon. SEM images were captured at different magnifications with a 5 kV acceleration voltage.

The particle size distributions and average diameters were ascertained by measuring more than 1000 randomly chosen capsules, utilizing ImageJ software v.1.54f (National Institute of Health, Bethesda, MD, USA). The particles’ area was measured and adjusted to the Beerkan estimation of soil transfer (BEST) model [30], as shown in Equation (1)
(1)$$F(D)={\left[1+{\left(\frac{D_{g}}{D}\right)}^{N}\right]}^{-M}$$
where *D* is the diameter of the particles (μm), *F*(*D*) is cumulative frequency associated with the diameters of the particles, *M* and *N* are two dimensionless shape parameters, and *D_g_* is a scale parameter (μm). The last three parameters were optimized to fit the data with the Solver MS Excel tool, minimizing the relative error of *F*.

#### 2.2.7. OEO Encapsulation Efficiency

To ascertain the amount of oil loaded into the capsules, measurements were taken for both the surface oil and the total oil contained within an equal quantity of capsules. The difference between these values allowed the determination of the oil present inside the capsules. Two similar procedures were followed to determine each of them.

An ethanol–sulfuric acid method was used [31] to obtain the calibration curves, and several concentrations of OEO in ethanol were prepared. The necessary quantity of the mother solution of oregano oil (0.25%v) in ethanol was taken, which, in a later step, was diluted in pure ethanol to reach the fixed concentration. A volume of 4 mL of 2 M HCl was added to this aliquot, and the mixture was heated for 30 min at 90 °C. After allowing the mixture to cool to room temperature, the rest of the ethanol was added to reach 2 mL. The mixture was shaken, transferred to a 15 mL Falcon tube, and centrifuged in a Hettich centrifuge (model UNIVERSAL 320R) for 5 min at 10,000× *g*. An absorbance scan was performed using a Varian Cary 100 Bio UV–vis spectrophotometer and a quartz cuvette to obtain the wavelength at which the absorption was at the maximum (279 nm).

To determine the surface oil of the capsules, 2 mL of pure ethanol was added to 10 mg of OEO-loaded particles by shaking gently to dissolve the surface oil on the capsules and centrifuging the mixture at 10,000× *g* for 5 min. The supernatant was taken, and 4 mL of 2 M HCl was added to it. It was heated for 30 min at 90 °C, cooled to room temperature, and measured in a spectrophotometer at 279 nm.

Finally, to determine the total oil, sulfuric acid was added to the same mass of capsules to dissolve the CS–GA capsules. The mixture was shaken and heated. After that, it was centrifuged. Then 2 mL of ethanol was added; the mixture was shaken again and measured in the spectrophotometer.

For both procedures, a blank was prepared with 10 mg of CS–GA nanoparticles to eliminate the influence of the particles on the calculations.

Finally, to calculate the encapsulation efficiency, Equation (2) was used
(2)$$Efficiency,\%=\frac{T-S}{T}\cdot 100$$
where T is the total mass (g) of OEO extracted from the capsules and S is the mass (g) of surface oil extracted without dissolving the capsules. The measurements were carried out in triplicate.

## 3. Results and Discussion

### 3.1. Formation of CS–GA Nanoparticles and Zeta Potential

The positive charge of chitosan (in an acid medium) [32,33,34] and the negative charge of Arabic gum (across almost the entire range of pH) [35] was clearly demonstrated.

On the one hand, at a low pH, chitosan was rich in protonated amino groups, which led to a positive zeta potential. Then, with the automatic addition of diluted NaOH, the pH increased and the zeta potential started decreasing because of the deprotonation of the amino groups of chitosan. From a pH slightly above 7, the zeta potential became approximately 0 and remained practically constant (Figure 1), in accordance with the literature [36].

On the other hand, Arabic gum, an anionic polysaccharide, showed a negative zeta potential throughout the entire range of pH evaluated due to the dissociation of the carboxyl groups present in its structure [37,38].

With the presence of acetic acid in the chitosan solution, there was a pH lower than 6, and the formation of the CS–GA complexes was more favorable; the difference between the values of zeta potential was greater, and their surfaces charges were the opposite and had a greater electrostatic attraction.

The CS–GA complexes showed an intermediate behavior between the two starting polymers, confirming the electrostatic interactions of the chitosan’s amino groups and the Arabic gum’s carboxylic groups. The trend was clear; the curves progressed from the values of chitosan to those of Arabic gum as the ratio of Arabic gum in the nanoparticles increased. The positive zeta potential of chitosan prevailed in the most acidic pH, especially at the CS–GA ratios of 3–1, 2–1, and 1–1, in which the measured values were greater than 30 mV; therefore, they were very stable suspensions. The CS–GA 1–2 suspension obtained a slightly lower zeta potential, but it was still at the value of around 20 mV in this pH range; therefore, the suspension was still stable. Lastly, the CS–GA 1–3 was the most unstable suspension; a greater proportion of Arabic gum worsened the stability of the particles at an acidic pH.

The same behavior was observed for the isoelectric point (IEP). When the amount of Arabic gum increased, the IEP decreased. The lowest value was obtained for Arabic gum at a pH of 2.21, in accordance with the literature [39]. Then the IEP increased in the pH range from 6 to 7, in the order of CS–GA 1–3, 1–2, 1–1, and 2–1, in the same way that the proportion of chitosan increased. The highest results for the IEP were obtained for CS–GA 3–1 and chitosan, with a similar value for both at a pH of around 7.7; similar results were obtained in the literature for chitosan [36,40,41].

### 3.2. Particle Size

In addition to the zeta potential, size is another fundamental parameter for understanding the movement that the particles make when emulsions are formed. The mobility of the particles is more limited when their size is larger, so particles that are not too large move rapidly towards the interface. This is justified by the Stokes model (Equation (3)) with which the Zetasizer calculates the diameter from the measurement of the diffusion coefficient
(3)$$d\left(H\right)=\frac{kT}{3\pi \eta D}$$
where *d*(*H*) is the hydrodynamic diameter, *D* is the translational diffusion coefficient, *k* is Boltzmann’s constant, *T* is the absolute temperature, and η is the dynamic viscosity. Therefore, larger sizes are obtained with low diffusivity values, thus limiting the movement of the particles.

In addition, the migration speed of the particles towards the interface was influenced by the interfacial tension, which will be discussed in the next section.

The samples of chitosan, Arabic gum, and mixtures with different ratios of the last two were measured. Each of the analyzed samples was prepared at a concentration of 2% wt, although for the measurement in the Zetasizer, the original solution was diluted 10 times.

From Table 1, it can be observed that the smallest particles in aqueous solution were Arabic gum nanoparticles (6.3 nm), which can be explained by the high solubility shown by Arabic gum in this study. In the case of chitosan, the value obtained under the conditions of measurement was 123.7 nm, 20 times higher than Arabic gum.

For the intermediate complexes, in all cases, larger particles were obtained, thus confirming the formation of hybrid particles via the complex coacervation of chitosan and Arabic gum. The compounds that were richer in chitosan achieved larger particle sizes.

### 3.3. Contact Angle

The formation of emulsions is favored by the presence of particles with intermediate wettability at the interface, i.e., with a contact angle close to 90°, evidencing a similar affinity for both phases. Figure 2 presents the contact angles of nanoparticles with different polymer ratios.

The contact angle of chitosan indicates its hydrophobic character [13]. As the zeta potential measurements showed, at a neutral pH, chitosan is insoluble, and this explains the obtained value of the contact angle.

Conversely, for gum Arabic, the contact angle of 70.2° indicated its greater hydrophilic nature; once the drop was deposited on the layer of gum Arabic particles, it was quickly diluted, and hence a lower contact angle was obtained.

In the remaining cases, with polymer mixtures at various ratios, contact angles of intermediate wettability were obtained, exhibiting a behavior that was contrary to that observed with isolated polymers; particles with a higher percentage of chitosan displayed more hydrophobic values, and vice versa.

For the formation of O/W Pickering emulsions, particles with a similar affinity for both phases are required, but with a slight hydrophilic character. The interface will curve, forming the phase in which the particles possess a greater surface area, namely the external phase of the emulsion [42].

For this reason, the most optimal particles for the formation of emulsions, considering the results obtained for the contact angle, were the CS–GA 1–2 nanoparticles, which exhibited a contact angle of 89.1°, just slightly below 90°. These particles demonstrated a slightly higher affinity for the aqueous phase and are therefore ideal for the formation and stabilization of O/W Pickering emulsions.

### 3.4. Interfacial Tension

For the interfacial tension, γ_OW_, the measurements are significant because they enable an understanding of how the particles migrate from the bulk of the aqueous solution to the interface. The closer the values are to zero, the less energy is required to disperse the internal phase and form the emulsion. The particles move to the interface, reducing the interfacial tension and facilitating stabilization of the emulsion.

The behavior of CS–GA nanoparticles at the interface can be modified in various ways. On the one hand, higher concentrations of nanoparticles in the aqueous solution normally yield lower values of interfacial tension [43]; a minimum number of particles is necessary to form a thin layer around the dispersed phase of the emulsion, avoiding the coalescency of the oil drops.

On the other hand, the polymer ratio (the chitosan–Arabic gum ratio) also affects the interfacial tension in the equilibrium value and imparts distinct behaviors to particles at the interface. Depending on the resulting final structure, they will exhibit greater or lesser ease in moving towards the interface, reducing the interfacial tension and facilitating the formation of stable Pickering emulsions.

The lowest value was obtained with a ratio of 1–2, which improved on the results obtained for 1–1 and 1–3.

#### 3.4.1. Influence of Nanoparticle Concentrations

To study the effect of particle concentrations on the values of interfacial tension, experiments were carried out using the CS–GA 1–2 particles, which obtained better results in the previous sections. Aqueous solutions of CS–GA nanoparticles ranging from 0.1% wt to 3% wt were used.

The experiments were conducted with a duration of 25 min, a sufficient time for the surface tension value to have stabilized, as observed in Figure 3. All of them were performed in triplicate.

In general, the particle concentration had a positive effect on reducing the interfacial tension at the water–oil interface. Figure 3 shows the evolution of the interfacial tension over the studied period. In all of the curves, either a more or less marked decrease in the interfacial tension from the initial value was observed.

As expected, increasing the concentration led to a decrease in the equilibrium value of γ_OW_. However, at the highest tested concentration (3% wt), the effect was not the same due to the increased solution viscosity and particle saturation in the aqueous solution, hindering the particles’ movement and not contributing to the rapid migration of particles to the interface. As a result, the achieved value was not improved compared with the 2% wt solution. Therefore, using a concentration of 2% wt was the optimal choice for forming Pickering emulsions.

#### 3.4.2. Influence of the Polymer Ratio

To study the effect of the polymer ratio, experiments were carried out using 2% wt particle solutions with different polymer ratios (Figure 4), which obtained the optimum results in the concentration study.

According to the results (Table 2), the lowest values were obtained by the chitosan particles (γ_OW_ = 2.89), and these results aligned with the assumptions of the contact angle and particle size experiments. The chitosan particles achieved the highest contact angle, indicating a more hydrophobic character and a greater tendency to migrate towards the organic phase. Additionally, they were also the largest particles, which, due to gravity, tended to descend towards the oil phase, further enhancing this observed behavior. Hence, if we solely focused on the values of interfacial tension, these particles would seem to be the most optimal for stabilization of the emulsion. However, it has been demonstrated that this is not the case.

A similar behavior was observed in the CS–GA particles that had a higher proportion of chitosan. Likewise, the contact angles deviated from 90°, but in this case, they had a more hydrophilic character, remaining within the internal part of the aqueous phase without positioning themselves at the interface and not promoting the stabilization of the O/W emulsion.

As for the particles with a higher proportion of Arabic gum, they attained higher values of equilibrium interfacial tension and exhibited a hydrophilic character.

Finally, the value achieved by the CS–GA 1–2 particles, which exhibited the best contact angle result, also yielded a relatively low value of interfacial tension. This indicated that these particles tended to migrate towards the interface, wrapping the oil droplets and consequently, contributing to the stabilization of the emulsion.

Thus, if we compare the overall results of the characterization studies of the different particles, the CS–GA 1–2 nanoparticles were the best option for forming O/W Pickering emulsions and were chosen to conduct the subsequent experiments for the encapsulation of oregano essential oil.

### 3.5. Biodegradability Test

GA is a natural plant polysaccharide, the oldest in use by humankind. Its use began around 5000 years ago [44]. The physicochemical and functional properties of GA may vary significantly depending on its source. Prasad et al. [45] conducted a review of the recent research on the production, processing, properties, application, and marketing of GA. The unique properties possessed by GA make it ideal for different industrial applications. Specifically, encapsulation has found a wide range of applications, in both the pharmaceutical and food industries. GA has been used as wall material for the encapsulation of essential oils using the coacervation technique [46] and isoflavones [47]. The extensive use of GA as a material for these purposes is due to its natural origin and, consequently, its biodegradable characteristics. In fact, Prasad et al. [45] described GA as biocompatible and biodegradable, and its source is renewable [47]. However, there are no data in the literature regarding the biodegradability of GA.

We tested the biodegradability of GA (Figure 5a), and found a biodegradation lower than 60%. Angelo et al. [48] studied the biodegradability of CS and found a final biodegradation of 59.9%. The addition of chitosan significantly increased the biodegradation of GA, making it more useful as a material for encapsulation (Figure 5b). Furthermore, the properties for the formation of Pickering emulsions were also enhanced.

The biocidal capacity of oregano essential oil has been extensively demonstrated and reported in the literature, having been applied for a wide range of uses.

Bayramoglu [49] showed that the essential oil of oregano had much stronger bactericidal activity than a commercial bactericide and fennel oil, making it useful in the leather industry. Eltai et al. [50] studied the efficacy of oregano oil as a biocide agent, measuring the reduction of genetically modified bioluminescent bacteria and yeast (*E. coli*, *P. aeruginosa*, and *S. aureus*). Gatti et al. [26] evaluated the biocidal activity of selected EOs against fungal and bacterial collections, and they concluded that oregano EO may be applied in the conservation of oil paintings, as it may potentially inhibit the growth of oil paintings’ biodeteriogens.

### 3.6. Encapsulation

Buchi experiments for two very different concentrations of OEO were carried out following the method described above. OEO emulsions of 10% wt and 0.5% wt were prepared to test how the concentration of oil influenced the formation of capsules.

#### 3.6.1. Characterization of Formed Capsules

By using the images captured with SEM (Figure 6), it was possible to ascertain the size and morphology of the capsules formed with each concentration of OEO.

Morphologically, both concentrations showed very similar appearances. Capsules tended to remain and form some aggregations; this could have been favored due to the deposition of the capsules on adhesive carbon tape.

By adjusting the particles’ diameter with Equation (1) to the BEST model described above, optimized parameters were achieved for both concentrations of OEO (Table 3). As observed in the SEM images, both exhibited very similar parameters, resulting in a narrower distribution for particles with 10% OEO compared with those with 5% OEO. The latter showed slightly more variation in particle size.

The greatest difference was observed in the D_g_ values, which provide an indication of the majority of the particles. A comparison of the values revealed that D_50_= 2.79 mm for 0.5%v and D_50_ = 2.45 mm for 10%v, with the width of the distribution being 1.23 mm and 1.86 mm, respectively, which signified a smaller size and less dispersion for particles with a higher concentration of oil, likely attributed to the potential aggregation of oil droplets during emulsification due to excess oil. The particles could not involve the aggregated droplets, and the surface oil increased after spray drying. Figure 7 shows particle size distributions for each concentration of OEO, where the lines correspond to the fit achieved using the BEST model.

Therefore, it is evident that the concentration of oil used in the preparation of the emulsion did not contribute significantly to the morphology and size of the capsules formed.

#### 3.6.2. OEO Encapsulation Efficiency

By following the described method, a calibration curve for the determination of the OEO concentration in the samples was obtained. The measures showed a high linearity with an R^2^ higher than 0.999 and a standard deviation that was, in all cases, lower than 0.02 in the determination of the absorbance of OEO %v. Thus, the method was valid for determining the concentration of oregano oil in the solution.

The total and surface oil for each concentration of OEO and a blank sample of the CS–GA 1–2 particles were measured to determine the %v OEO present in each sample.

Through calculations using Equation (2), disparities were identified between the two different concentrations of OEO (%v) used for the formation of the emulsion. The capsules prepared with 0.5%v OEO achieved an encapsulation efficiency of 60.1%, as outlined in Equation (1), while those prepared with 10%v achieved 36.4%.

Thus, as indicated by the results, it is evident that significantly increasing the oil concentration did not enhance the efficiency of encapsulation. This may be attributed to the presence of numerous oil droplets during the formation of the emulsion, leading to coalescence, which hindered the complete coating of oil droplets by CS–GA particles.

Figure 8 shows a schematic diagram with the main optimized parameters.

## 4. Conclusions

Capsules comprised of chitosan–Arabic gum nanoparticle complexes were successfully synthesized through spray-drying. To optimize the parameters of preparing the emulsion, the particle size, zeta potential, contact angle, and interfacial tension were evaluated. The CS–GA 1–2 nanoparticles reached a size of 257.4 nm after complex coacervation when conducted under acidic pH conditions, generating a distinct zeta potential between the two polymers. The measured contact angle was 89.1 degrees, indicating their suitability for stabilizing oil-in-water (O/W) emulsions. This was due to their hydrophilic properties, which caused the particles to predominantly gather at the interface and effectively encapsulate the oil droplets. A 2% wt concentration of these particles notably reduced the equilibrium interfacial tension to 7.72 mN/m, and facilitated particles’ migration to the interface.

Additionally, the inclusion of chitosan enhanced the biodegradability of Arabic gum nanoparticles, fulfilling expectations. The capsules demonstrated effective encapsulation of oregano oil through atomization-based drying. The concentration of oil did not directly impact the particles’ formation, and maintained capsule sizes between 2 and 3 μm. However, capsules prepared with a 5%v concentration of OEO achieved a 60% oil encapsulation efficiency, outperforming the outcomes of higher oil concentrations that increased the surface oil content.

These capsules, leveraging the properties of chitosan and Arabic gum, as well as oregano oil, offer remarkable versatility for various applications. Their potential relevance in biodetergent formulations is particularly noteworthy, and will be explored in future research endeavors.

## Figures and Tables

**Figure 1 nanomaterials-13-02651-f001:**
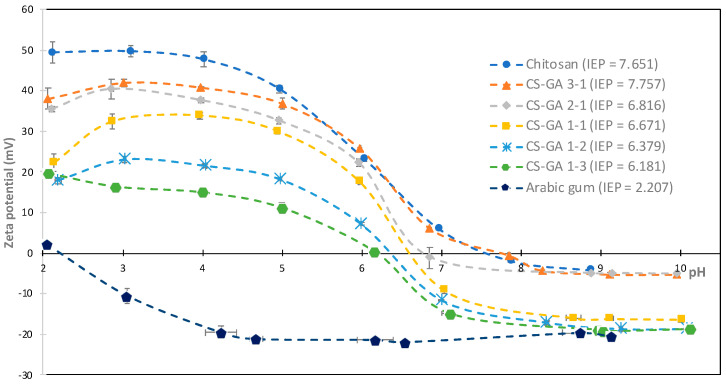
Titration curves of chitosan, Arabic gum, and CS–GA complexes.

**Figure 2 nanomaterials-13-02651-f002:**
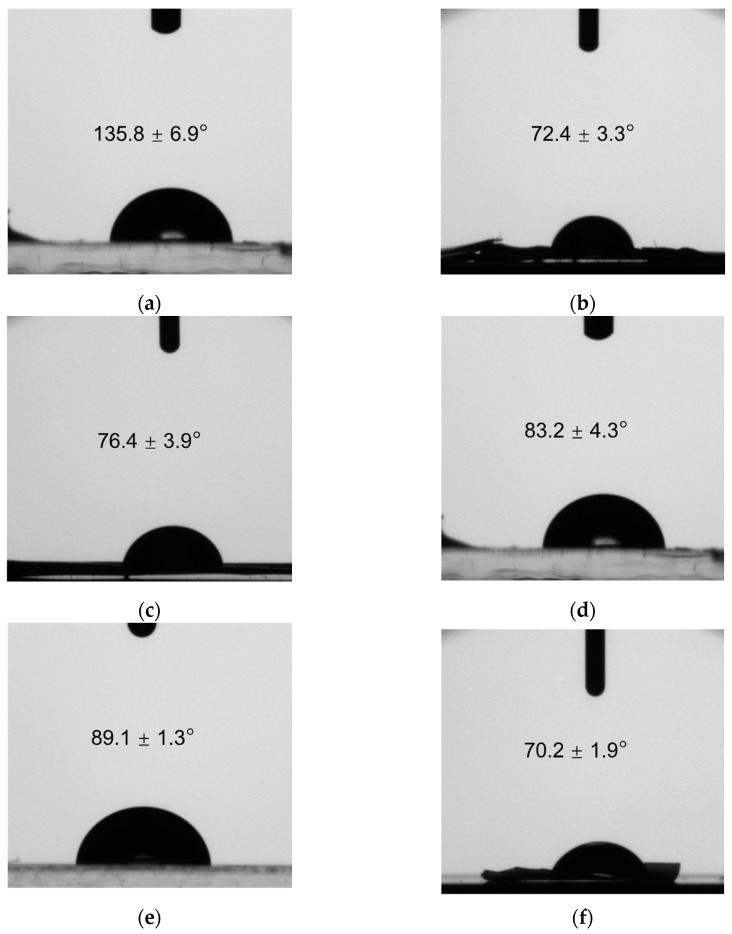
Contact angles for different nanoparticle compositions. (**a**) Chitosan; (**b**) CS–GA 3–1; (**c**) CS–GA 2–1; (**d**) CS–GA 1–1; (**e**) CS–GA 1–2; (**f**) Arabic gum.

**Figure 3 nanomaterials-13-02651-f003:**
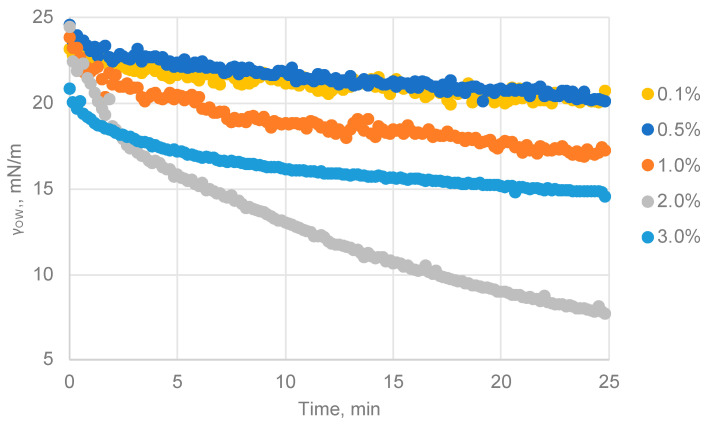
Measurements of interfacial tension over time for different concentrations (% wt) of CS–GA 1–2 nanoparticles.

**Figure 4 nanomaterials-13-02651-f004:**
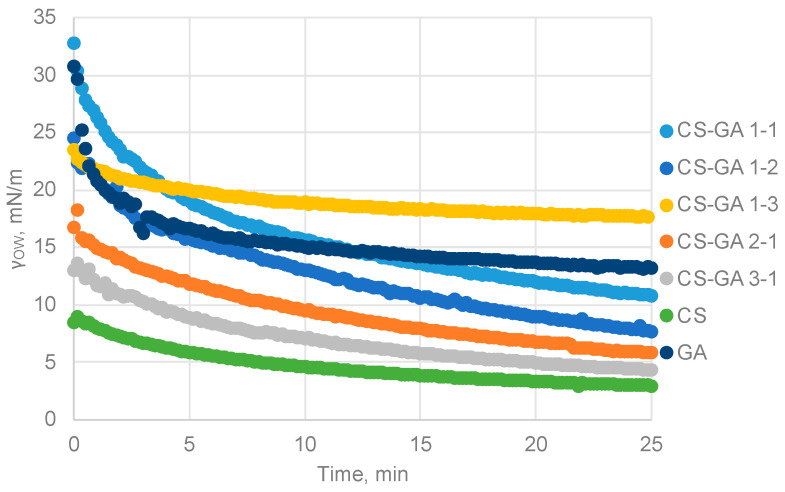
Interfacial tension values obtained for different CS–GA polymer ratios.

**Figure 5 nanomaterials-13-02651-f005:**
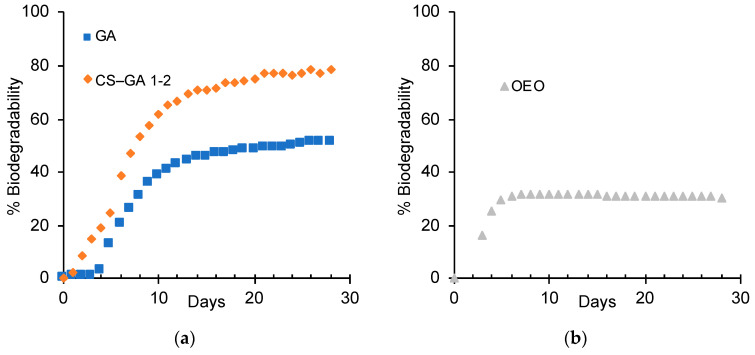
Biodegradability test curves for (**a**) GA and CS–GA 1–2, and (**b**) OEO.

**Figure 6 nanomaterials-13-02651-f006:**
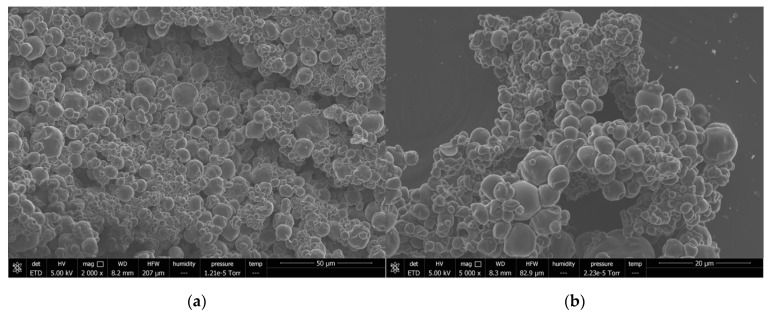
SEM images of the formed capsules: (**a**–**c**) 10% wt OEO and (**d**–**f**) 0.5% wt OEO. The horizontal field widths (HFWs) were (**a**) 207 μm, (**b**) 82.9 μm, (**c**) 41.4 μm, (**d**) 207 μm, (**e**) 82.9 μm, and (**f**) 104 μm. Scale bar indicates (**a**,**d**) 50 μm, (**b**,**e**,**f**) 20 μm, and (**c**), 10 μm.

**Figure 7 nanomaterials-13-02651-f007:**
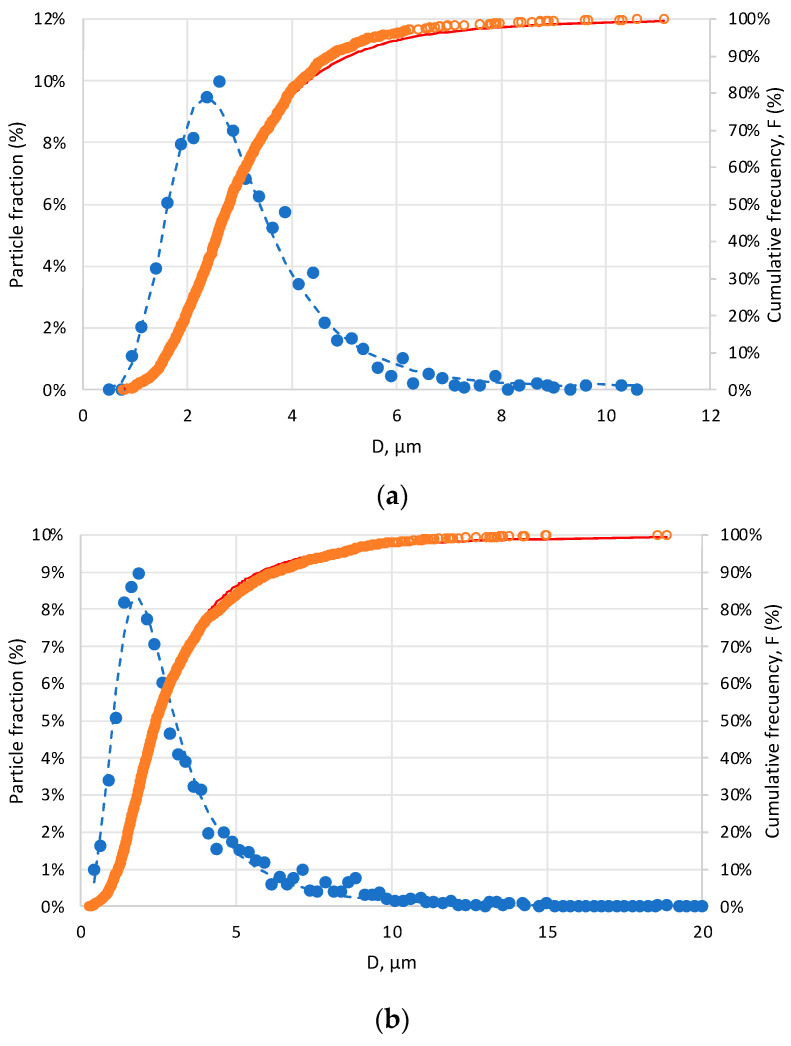
Adjustments to the BEST model and distributions of particle diameter: (**a**) 0.5%v OEO and (**b**) 10%v OEO. Blue marker: particle size fraction; orange marker: cumulative frequency.

**Figure 8 nanomaterials-13-02651-f008:**
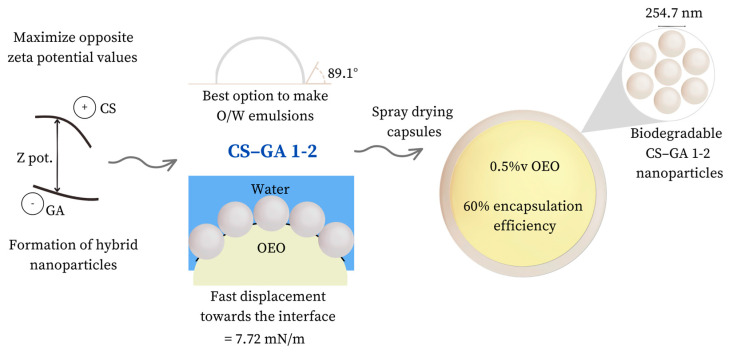
Graphical conclusions and schematic optimization of the variables and processes.

**Table 1 nanomaterials-13-02651-t001:** Particle sizes for different polymer (CS–GA) ratios.

Polymer Ratio	Size, nm	Std. Dev., nm
CS	123.7	±4.0
3–1	269.5	±5.6
2–1	256.2	±2.7
1–1	237.1	±3.8
1–2	257.4	±1.7
1–3	134.6	±4.5
GA	6.3	±0.6

**Table 2 nanomaterials-13-02651-t002:** Interfacial results for nanoparticles with different polymer ratios.

Polymer Ratio	γ_OW-MIN_, mN/m	Std. Dev., mN/m
CS	2.89	±0.14
3–1	4.39	±0.34
2–1	5.88	±0.29
1–1	10.82	±0.28
1–2	7.72	±0.17
1–3	17.68	±0.09
GA	13.13	±0.52

**Table 3 nanomaterials-13-02651-t003:** Optimized parameters for the BEST model, Equation (1).

%v OEO	N	M	Dg, µm
0.5%	3.51	1.41	2.45
10%	2.47	1.45	2.02

## Data Availability

The data presented in this study are available on request from the corresponding author.

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
