# Peer review of "Synthesis and Characterization of Environmentally Friendly Chitosan–Arabic Gum Nanoparticles for Encapsulation of Oregano Essential Oil in Pickering Emulsion"

_nanomaterials, 2023, doi:10.3390/nano13192651_

Round 1

Reviewer 1 Report

This manuscript describes the use of combined chitosan-arabic gum nanoparticles for obtaining capsules comprising oregano seed oil by the Pickering emulsion technique followed by spray-drying. The method is well known, as are the benefits of the essential oil, so there is limited novelty in this sense. Perhaps, the combination of chitosan with arabic gum to obtain the delivery system might be of interest to Nanomaterials readership. 

Issues that need to be addressed during the revision round(s) to follow:

Abstract: many of the acronyms used here are not defined. They should be defined here, as this is the time of their first occurrence in the manuscript.

Lines 62-64: All these claims must be backed up by citing relevant references

The whole Materials and Methods section must be revised. There is a complete lack of consistency in verb tenses, which are switched randomly between past, present and even future tense. Future tense should definitely not be used, as this is work done, not future goals. Also, past tense is generally considered more appropriate than present. The same tense inconsistency happens again later in section 3.6.2

You must add an additional sub-section to Materials and Methods about Chemicals used: list vendors and purities for all chemicals used in all your experimental protocols

Line 169: Mention the brand and type of centrifuge and spectrophotometer. Mention also g-force for centrifuge as 9000 rpm is not the same for all centrifuges, it depends on g-force.

Equation 2: mention the units of measurement for T and S

Line 188: "As it says in bibliography" - this is very unprofessional scientific writing style

Line 197: present (not "presented")

Line 202: the word "Figure" does not belong here, on this line

Line 228: there is a blank here; the symbol for dynamic viscosity is missing

Line 236: particles ... were ... (plural noun, plural verb needed)

Table 1: the Std.Dev. has units of nm also

Line 255: Full stop after citation [12] is needed. Then start a new sentence. There are several other occurrences in the manuscript where this happens, i.e. clearly different sentences are separated by commas rather than full stops.

Lines 282-283: Rephrase to "a minimum amount of particles is necessary to ..."

There should be a main heading 3.4 prior to sub-heading 3.4.1

Figure 3 is shown twice

The figure caption of Figure 3 should explain clearly what type of curves are those shown (i.e. what is plotted against what) and what the % values shown in the legend mean.

Throughout the manuscript: polymer ratio (not "polymers" ratio)

Table 1: title must be revised; include also the standard deviations for the interfacial tensions shown here as previously done for other parameters

Figure 5 (a): why isn't also CS alone shown ? This is also a necessary control for proper comparison.

What are the differences between figures 6a, 6b and 6c and respectively between figures 6a, 6e and 6f? Scale bars and other bottom information are impossible to read in these figures, fonts are illegible.

Table 2: title is copy-pasted from Table 1. It is incorrect! Give a proper title and make sure you explain the notations used in the heading of all columns.

Line 403: avoid using contracted forms like "couldn't" in scientific writing

Line 418: determining (not "determine")

Line 420: present (not "presented")

Line 422-424: Poor English, the message of this phrase is not clear. Revise!

Line 445: "m"? (i.e. meters?) I think not.

Acknowledgements: This is simply the standard text copied. Delete this section if you have no one to acknowledge.

References are not formatted in the style demanded by this journal. Reformatting is needed!

Line 361: insert a proper reference using your citation manager, not a side comment of the pdf file.

English is subpar for the most part. The authors are urged to seek professional help with English editing of their revised manuscript.

Author Response

Dear Reviewer,

Please, find enclosed a revision of our manuscript entitled “Synthesis and characterization of environmentally friendly chitosan-arabic gum nanoparticles for encapsulation of oregano essential oil in Pickering emulsion”. (nanomaterials-2615098). We would like to thank the Editorial committee for giving us the opportunity to improve our manuscript, and the reviewers for their thoughtful and constructive comments. We have considered all the suggestions and have incorporated them into the revised manuscript. Changes to the previous manuscript have been marked using Track Changes, and we believe our manuscript is stronger as a result of these modifications. An itemized point-by-point response to the reviewers’ comments is presented below.

(Pagination of the answers are referred to the revised manuscript).

Reviewer #1:

The authors systematically explored the optimization of emulsions conditions to obtain the best performance of spray drying to make chitosan-arabic gum hybrid capsules with oregano essential oil in the core. However, there are many areas in the main text thrown in that need to be revised by the author before the manuscript is published.

First and foremost, it should be noted that all recommendations regarding format changes and editorial improvements have been adhered to throughout the entire article.

  1. Abstract: many of the acronyms used here are not defined. They should be defined here, as this is the time of their first occurrence in the manuscript.

Response:

We have reviewed the entire abstract and all the acronyms have been put in the correct place, at the first time that the words appear in the text.

  1. Lines 62-64: All these claims must be backed up by citing relevant references

Response:

Following the guidance of the reviewer some references have been added:

  • Leyva-López, N.; Gutiérrez-Grijalva, E. P.; Vazquez-Olivo, G.; Heredia, J. B. Essential Oils of Oregano: Biological Activity beyond Their Antimicrobial Properties. Molecules 2017, 22(6), 989. https://doi.org/10.3390/molecules22060989
  • Gatti, L; Troiano, F; Vacchini, V; Cappitelli, F.; Balloi, A. An In Vitro Evaluation of the Biocidal Effect of Oregano and Cloves’ Volatile Compounds against Microorganisms Colonizing an Oil Painting—A Pioneer Study. Applied Sci. 2021, 11(1), 78. https://doi.org/10.3390/APP11010078.
  • Rhoades, J.; Gialagkolidou, K.; Gogou, M.; Mavridou, O.; Blatsiotis, N.; Ritzoulis, C.; Likotrafiti, E. Oregano Essential Oil as an Antimicrobial Additive to Detergent for Hand Washing and Food Contact Surface Cleaning. J. Appl. Microbiol. 2013, 115(4), 987–994. https://doi.org/10.1111/jam.12302

  1. The whole Materials and Methods section must be revised. There is a complete lack of consistency in verb tenses, which are switched randomly between past, present and even future tense. Future tense should definitely not be used, as this is work done, not future goals. Also, past tense is generally considered more appropriate than present. The same tense inconsistency happens again later in section 3.6.2.

Response:

The entire text has been reviewed to select the most appropriate verb tenses for use throughout the entire article. Furthermore, to address this point, the English language has been revised by the Language Editing Services of MDPI.

  1. You must add an additional sub-section to Materials and Methods about Chemicals used: list vendors and purities for all chemicals used in all your experimental protocols

Response:

The recommended information by the reviewer has been incorporated for all chemicals through lines 83 to 91. The previous “2.1 Materials” sub-section has been changed by “2.1 Chemicals” sub-section

  1. Line 169: Mention the brand and type of centrifuge and spectrophotometer. Mention also g-force for centrifuge as 9000 rpm is not the same for all centrifuges, it depends on g-force.

Response:

The model of centrifuge is added in line 183. The equivalent g-force has been included in place of rpm.

  1. Equation 2: mention the units of measurement for T and S

Response:

In Equation 2, T and S were used to calculate encapsulate oil efficiency. These refer to the total and surface oil of oregano essential oil present in the capsules formed through spray drying. Therefore, they serve for calculation if they are in the same units; thus, the total mass (T) and the surface oil (S) extracted are expressed in grams.

  1. Line 188: "As it says in bibliography" - this is very unprofessional scientific writing style

Response:

The sentence has been rewritten. Line 203

  1. Line 197: present (not "presented")

Response:

Corrected.

  1. Line 202: the word "Figure" does not belong here, on this line

Response:

Corrected.

  1. Line 228: there is a blank here; the symbol for dynamic viscosity is missing

Response:

Corrected.

  1. Line 236: particles ... were ... (plural noun, plural verb needed)

Response:

Corrected.

  1. Table 1: the Std.Dev. has units of nm also

Response:

The units for the Std.Dev. have been included in Table 1.

  1. Line 255: Full stop after citation [12] is needed. Then start a new sentence. There are several other occurrences in the manuscript where this happens, i.e. clearly different sentences are separated by commas rather than full stops.

Response:

Corrected.

  1. Lines 282-283: Rephrase to "a minimum amount of particles is necessary to ..."

Response:

Corrected.

  1. There should be a main heading 3.4 prior to sub-heading 3.4.1

Response:

Thank you for the comment, but there already was a main heading 3.4. Interfacial tension before 3.4.1 subsection.

  1. Figure 3 is shown twice

Response:

Corrected.

  1. The figure caption of Figure 3 should explain clearly what type of curves are those shown (i.e. what is plotted against what) and what the % values shown in the legend mean.

Response:

Thank you to the reviewer for the feedback. As a result, the figure caption has been rewritten to provide a clearer explanation of the information displayed in the graph. It has been modified as follows: "Measurement of interfacial tension over time for different concentrations (% wt) of CSGA 1-2 nanoparticles.”

  1. Throughout the manuscript: polymer ratio (not "polymers" ratio)

Response:

Corrected.

  1. Table 1: title must be revised; include also the standard deviations for the interfacial tensions shown here as previously done for other parameters

Response:

The title of Table 2 has been revised and Std.Dev. have been included.

  1. Figure 5 (a): why isn't also CS alone shown? This is also a necessary control for proper comparison.

Response:

The most relevant biodegradability data are the biodegradation achieved by the CS-GA system proposed in this work. Our first objective was to improve the GA biodegradability. We have included a value of the CS biodegradability by other authors.

  1. What are the differences between figures 6a, 6b and 6c and respectively between figures 6a, 6e and 6f? Scale bars and other bottom information are impossible to read in these figures, fonts are illegible.

Response:

The differences between images are the magnification and capsules details. We modified the caption of Figure 6 to include additional information about the horizontal field width and scale bars.

  1. Table 2: title is copy-pasted from Table 1. It is incorrect! Give a proper title and make sure you explain the notations used in the heading of all columns.

Response:

We apologize for the error made in indicating the title of the table. This mistake may have occurred during the transcription of the article to ensure proper formatting. As a result, the correct title for the table has been added, clearly indicating the information presented in the table.

  1. Line 403: avoid using contracted forms like "couldn't" in scientific writing

Response:

Corrected.

  1. Line 418: determining (not "determine")

Response:

Corrected.

  1. Line 420: present (not "presented")

Response:

Corrected.

  1. Line 422-424: Poor English, the message of this phrase is not clear. Revise!

Response:

The idea presented in the paragraph has been rewritten to enhance the clarity of the message. In addition, the manuscript has been revised by the Language Editorial Services.

  1. Line 445: "m"? (i.e. meters?) I think not.

Response:

There was an error in the transcription of the units of measurement for the size of the formed capsules, and therefore, “m” has been changed by “μm”.

  1. Acknowledgements: This is simply the standard text copied. Delete this section if you have no one to acknowledge.

Response:

Done

  1. References are not formatted in the style demanded by this journal. Reformatting is needed!

Response:

All the references have been revised according to the style of the journal.

  1. Line 361: insert a proper reference using your citation manager, not a side comment of the pdf file.

Response:

The mistake has been corrected.

Reviewer 2 Report

This article is devoted to obtaining chitosan-gum arabic nanoparticles for encapsulation of oregano essential oil. The article is written in an understandable language, the main ideas and results are not in doubt. In terms of volume, subject and level of research, this work meets the requirements of the Journal. The relevance of research is due to the growing need for eco-friendly materials and methods. There are some points that need to be improved:

1. Abstract can be expanded.

2. It is desirable to more clearly define the purpose of the study.

3. What is the resistance of these nanoparticles to the acidity of the medium? how stable are they?

4. What is the practical significance of this study?

5. It is desirable to cite: 10.1134/S1560090420020050.

6. It is desirable to make the conclusions more concise.

Author Response

Dear Reviewer,

Please, find enclosed a revision of our manuscript entitled “Synthesis and characterization of environmentally friendly   chitosan-arabic gum nanoparticles for encapsulation of oregano essential oil in Pickering emulsion”. (nanomaterials-2615098). We would like to thank the Editorial committee for giving us the opportunity to improve our manuscript, and the reviewers for their thoughtful and constructive comments. We have considered all the suggestions and have incorporated them into the revised manuscript. Changes to the previous manuscript have been marked using Track Changes, and we believe our manuscript is stronger as a result of these modifications. An itemized point-by-point response to the reviewers’ comments is presented below.

(Pagination of the answers are referred to the revised manuscript).

Reviewer #2:

This article is devoted to obtaining chitosan-gum arabic nanoparticles for encapsulation of oregano essential oil. The article is written in an understandable language, the main ideas and results are not in doubt. In terms of volume, subject and level of research, this work meets the requirements of the Journal. The relevance of research is due to the growing need for eco-friendly materials and methods. There are some points that need to be improved:

  1. Abstract can be expanded.

Response:

According to the reviewer’s comments the abstract has been expanded.

  1. It is desirable to more clearly define the purpose of the study.

Response:

To clarify the purpose of the study we have added an additional explanation in the last paragraph of the introduction, lines 73-80.

  1. What is the resistance of these nanoparticles to the acidity of the medium? how stable are they?

Response:

So far, we have only tested the resistance of capsules in a hard acid medium, H2SO4 used to dissolve the capsules during the encapsulation efficiency test. Nevertheless, we plan to study the stability of the capsules in the whole range of conditions.

  1. What is the practical significance of this study?

Response:

As has been indicated in the corrected last paragraph of the introduction (lines 73-80), the practical significance is to define a process and the optimal conditions to obtain environmentally friendly capsules made by chitosan and gum Arabic nanoparticles, for their inclusion in detergent formulations and other applications as food technology.

  1. It is desirable to cite: 10.1134/S1560090420020050.

Response:

Thank you for the suggestion, the reference has been added in the introduction section as reference [15] (line 61).

  1. Slyusarenko, N.V.; Vasilyeva, N.Y.; Kazachenko, A.S. Gerasimova, M.A.; Romanchenko, A.S.; Slyusareva, E. A. Synthesis and Properties of Interpolymer Complexes Based on Chitosan and Sulfated Arabinogalactan. Polymer Science, Serie B 62, 272–278 (2020). https://doi.org/10.1134/S1560090420020050

  1. It is desirable to make the conclusions more concise.

Response:

The conclusions have been rewritten in a more concise way.

Reviewer 3 Report

1.        Title should be modified as “Synthesis and characterization of chitosan-arabic gum nanoparticles for encapsulation of oregano essential oil in pickering emulsion for detergent formulation”.

2.        Another keyword “detergent formulation” should be included in the keywords section.

3.        All the purchase details of chemicals/reagents and instruments/equipment/software/kits should be provided as state, city, and country in the case of USA as well as city and country in the case of other countries. Also, for the second instance of the same vendor/company’s mention, the authors can simply mention the company name, for instance as Sigma-Aldrich and not very time Sigma-Aldrich (USA).

4.        In the section, one reference citation each should be provided for procedures described in Section 2.

5.        There are too many small 1 or 2-sentence paragraphs throughout the manuscript, which need to be combined to have a few large paragraphs.

6.        In all the tables ensure that the abbreviations and variables used are provided in the full form in the respective table’s footnotes, while those used in figures are explained in the respective figure’s caption.

7.        Rewrite the Figure 6 caption for clarity.

8.        The secondary y-axis is not clear in Figure 7.

9.        A schematic diagram, showing the optimized parameters and their optimized values to provide the readers with the take-home points, should be provided just above the conclusion.

10.     Double check if all the units are provided correctly and abbreviated properly, for example, ‘milliliters’ as mL” and so on.

Minor editing of English language required

Author Response

Dear Reviewer,

Please, find enclosed a revision of our manuscript entitled “Synthesis and characterization of environmentally friendly chitosan-arabic gum nanoparticles for encapsulation of oregano essential oil in Pickering emulsion”, (nanomaterials-2615098). We would like to thank the Editorial committee for giving us the opportunity to improve our manuscript, and the reviewers for their thoughtful and constructive comments. We have considered all the suggestions and have incorporated them into the revised manuscript. Changes to the previous manuscript have been marked using Track Changes, and we believe our manuscript is stronger as a result of these modifications. An itemized point-by-point response to the reviewers’ comments is presented below.

(Pagination of the answers is referred to the revised manuscript).

Reviewer #3:

  1. Title should be modified as “Synthesis and characterization of chitosan-arabic gum nanoparticles for encapsulation of oregano essential oil in pickering emulsion for detergent formulation”.

Response:

The authors appreciate the suggestion for the modification of the title. We have changed the paper title to: “Synthesis and characterization of environmentally friendly chitosan-Arabic gum nanoparticles for encapsulation of oregano essential oil in Pickering emulsion”. Nevertheless, we think that the application of these capsules could not be exclusive for detergent formulation and therefore “for detergent formulation” has not been included in the title, since in the article the nanoparticles obtained have not been tested in a detergent formulation.

  1. Another keyword “detergent formulation” should be included in the keywords section.

Response:

Done

  1. All the purchase details of chemicals/reagents and instruments/equipment/software/kits should be provided as state, city, and country in the case of USA as well as city and country in the case of other countries. Also, for the second instance of the same vendor/company’s mention, the authors can simply mention the company name, for instance as Sigma-Aldrich and not very time Sigma-Aldrich (USA).

Response:

According to the reviewer’s comment, the purchase details of the chemical used in the study have been added in section 2.1 Chemicals, lines 83-91.

  1. In the section, one reference citation each should be provided for procedures described in Section 2.

Response:

According to the reviewer’s comments references [28] and [29] have been included:

  1. Bordi, F.; Chronopoulou, L.; Palocci, C.; Bomboi, F.; Di Martino, A.; Cifani, N.; Pompili, B.; Ascenzioni, F.; Sennato, S. Chitosan-DNA Complexes: Effect of Molecular Parameters on the Efficiency of Delivery. Colloids Surf A Physicochem Eng Asp 2014, 460, 184–190, doi:10.1016/j.colsurfa.2013.12.022.
  2. Jones, O.G.; McClements, D.J. Functional Biopolymer Particles: Design, Fabrication, and Applications. Compr Rev Food Sci Food Saf 2010, 9, 374–397.

  1. There are too many small 1 or 2-sentence paragraphs throughout the manuscript, which need to be combined to have a few large paragraphs.

Response:

English review by the editorial service has been done.

  1. In all the tables ensure that the abbreviations and variables used are provided in the full form in the respective table’s footnotes, while those used in figures are explained in the respective figure’s caption.

Response:

According to the reviewer´s comment, all the tables and figure’s captions have been revised.

  1. Rewrite the Figure 6 caption for clarity.

Response:

According to the reviewer´s comment, the caption of Figure 6 has been rewritten adding more information.

  1. The secondary y-axis is not clear in Figure 7.

Response:

Corrected.

  1. A schematic diagram, showing the optimized parameters and their optimized values to provide the readers with the take-home points, should be provided just above the conclusion.

Response:

According to the reviewer's comment, a schematic diagram with the optimized parameters has been included above the conclusion.

  1. Double check if all the units are provided correctly and abbreviated properly, for example, ‘milliliters’ as mL” and so on.

Response:

According to the reviewer’s comment, the units have been reviewed and corrected.

Round 2

Reviewer 1 Report

Please provide units for all parameters that appear in equation 1 also for better clarity.

Author Response

Dear Reviewer,

Thank you once again for your suggestion.

  1. Please provide units for all parameters that appear in equation 1 also for better clarity.

We have included units for the parameters D and Dg (μm), and we have also indicated that the parameters M and N are dimensionless (lines 165-167).

Reviewer 3 Report

After carefully reviewing the reviewers' comments and authors' responses, I recommend acceptance of this article for publication in Nanomaterials, as the authors have satisfactorily addressed all the comments raised by reviewers and substantially improved the overall quality of the article.

 Minor editing of English language required

Author Response

Dear Reviewer,

Thank you once again for your suggestion.

  1.  Minor editing of English language required

Response:

The manuscript has already been revised by MDPI's Editorial English Language team. Additionally, we have made minor English language changes in the revised version.